# Towards an Acoustic Geometry in Slightly Viscous Fluids

Mayank Pathak [1] and Parthasarathi Majumdar [2,*]

[1]  Department of Physics, Indian Institute of Science Education and Research, Pune 411008, India; mayank.pathak@students.iiserpune.ac.in
[2]  School of Physical Sciences, Indian Association for the Cultivation of Science, Kolkata 700032, India
*  Correspondence: parthasarathi.majumdar@iacs.res.in

**Abstract:** We explore the behaviour of barotropic and irrotational fluids with a small viscosity under the effect of first-order acoustic perturbations. We discuss, following the extant literature, the difficulties in gleaning an acoustic geometry in the presence of viscosity. In order to obviate various technical encumbrances, when viscosity is present, for an extraction of a possible acoustic geometry, we adopted a method of double perturbations, whereby dynamical quantities such as the velocity field and potential undergo a perturbation both in viscosity and in an external acoustic stimulus. The resulting perturbation equations yield a solution which can be interpreted in terms of a generalised acoustic geometry, over and above the one known for inviscid fluids.

**Keywords:** kinematic viscosity; double perturbation; acoustic geometry; lorentz symmetry breakdown

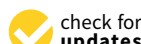



## 1. Introduction

Hawking radiation from black holes is theoretically an extremely plausible and compelling phenomenon. However, this phenomenon is astrophysically unobservable from any stellar black hole observed through its accretion disc emission. This is due to the fact that the Hawking temperature for a stellar black hole of a mass of the order of a solar mass is a fraction of a degree Kelvin. Thus, any astrophysical signal of Hawking radiation from such a black hole is *very likely swamped* by the 3° Kelvin cosmic microwave background (CMB). Similarly, the Penrose process of energy extraction from a rotating black hole, and its wave analogue superradiance, while remarkably compelling, have never been detected observationally in emissions from accretion discs of the spinning stellar black holes which have been studied extensively. The reason here is the same: if the superradiant signal from a black hole has a thermal spectrum, its equilibrium temperature can never exceed the CMB temperature. If this signal has a coherent component, unless it is very intense, it is likely to thermalise and be swamped again by the CMB. Another phenomenon occurring in the spacetime of spinning black holes is the dragging of inertial frames leading to the Lense–Thirring precession of spinning text gyroscopes in those spacetimes. This phenomenon has actually been observed, not for a black hole, but in the weakly curved spacetime around the earth—by Gravity Probe B [1]. However, in strong gravity situations, as in black hole spacetimes, this phenomenon is still beyond observational accessibility. Note that the phenomena mentioned occur in spacetimes of fixed *non-dynamical* geometry.

This latter fact was brilliantly adapted by W. Unruh [2,3] in 1981 as an acoustic analogue in inviscid, barotropic fluids, to demonstrate the possible observability of acoustic Hawking radiation of phonons from black hole analogues corresponding to acoustic perturbation of specific fluid flows. Likewise, acoustic superradiance [4] and Lense–Thirring precession [5,6] have also been shown to be within the realm of observability for rotating acoustic black hole analogues. More recently, definitive reports of actual observation of acoustic Hawking radiation [7] (see also [8]) and acoustic superradiance [9] have lent credence to this entire effort.

The acoustic analogy is defined for first-order acoustic perturbations propagating on barotropic, irrotational and inviscid background fluid flows. Such perturbations appear to simulate a scalar field propagating in a curved (acoustic) spacetime background with a Lorentzian signature. For quantum fluids such as superfluid helium or indeed a Bose–Einstein condensate in the hydrodynamic approximation, the inviscidity is never an issue. However, for real fluids like water, the viscosity is a property that cannot be ignored. Visser, in his 1998 review [3] of analogue gravity, showed that the direct introduction of viscosity in these systems leads to a violation of the Lorentz symmetry. This implies that an acoustic geometry cannot be demonstrated as simply for viscous fluid systems under acoustic perturbations. However, it might be worthwhile to study slightly viscous fluids and treat their very small viscosity as a perturbation while considering the flow of the fluid. This can also help in developing more insight into the existence of an acoustic metric in viscous backgrounds.

Torres et al. [9] have demonstrated the existence of superradiance resulting from acoustic perturbations using water as the background for their experiment. Although small, water has a non-zero viscosity, and this needs to be taken into account when working on such phenomena vis-a-vis the acoustic analogy. The reported observation and verification of the phenomenon of superradiance in water provided the motivation for exploring the possibility of an acoustic metric in slightly viscous fluids by introducing the viscosity perturbatively.

In the present paper, we investigate the implications of introducing viscosity perturbatively in a barotropic and irrotational fluid and calculate the viscosity-perturbed acoustic metric. The rest of the paper is arranged as follows: Section 2 contains a brief summary of the existing literature on acoustic analogue gravity and points out the gaps for further investigation. In Section 3, we describe the double perturbation method and derive the necessary equations. Using these equations, we obtain, in Section 4, the viscosity perturbation correction to the symmetric rank 2 tensor $f^{\alpha\beta}$ discussed in [3] for inviscid fluids. The corresponding acoustic metric in slightly viscous fluids, using these viscosity perturbation corrections. In this section, we also focus on slightly viscous fluids in two space dimensions and derive the acoustic metric in (2+1)-dimensional vortex-type flow in viscous fluids. Our results, along with the outlines of future work, are presented in Section 5.

A remark as a disclaimer: The entire field of 'analogue gravity' is primarily intended to provide experimental and observational evidence, as an analogy, for proposed and eminently plausible phenomena in physical spacetime physics which are beyond observational accessibility. No practitioner in this field of activity ever considers this as a *replacement* of actual gravitational physics. However, the use of analogies in theoretical physics continues unabated because analogies provide approaches which may circumvent seemingly unassailable obstacles in theory and/or experiment. In astrophysics, the analogy of pulsar emissions from rotating neutron stars, with the light emitted from a lighthouse, has been rather useful in constructing a theory of pulsar emissions when neutron stars have strong magnetic fields. Theories of origins of interstellar and intergalactic magnetic fields, based on an analogy with an electrical dynamo, have considerably aided in understanding where such magnetic fields may originate from. In elementary physics, the analogy between a damped harmonic oscillator and an electrical LCR circuit is often the basis of textbook treatments of circuit theory. Thus, analogies often open up new ways of looking at those parts of physics where understanding and/or observational evidence is sparse.

## 2. Essentials of Acoustic Gravity Analogy

The basic framework of acoustic analogue gravity for first-order acoustic perturbations in barotropic and irrotational fluids in both the inviscid and viscous case is briefly reviewed in this section, as a motivation for the subsequent sections.

### 2.1. Auxiliary Fluid Mechanics

The fluid mechanics equations [10,11] governing viscous fluid systems are given by

$$
\begin{aligned}
\text{Continuity equation}: \partial_t \rho + \nabla \cdot (\rho \vec{v}) &= 0 \\
\text{Navier} - \text{Stokes equation}: \rho(\partial_t \vec{v} + \vec{v} \cdot \nabla \vec{v}) &= -\nabla p - \rho \nabla \psi - \rho \nabla \Psi \\
&+ \eta \nabla^2 \vec{v} + (\zeta + \frac{1}{3}\eta)\nabla(\nabla \cdot \vec{v})
\end{aligned}
\tag{1}
$$

Here $\rho$ and $\vec{v}$ are the density and flow velocity of the fluid, respectively, and $\zeta$ and $\eta$ are the two viscosity coefficients. The terms present on the RHS of the Euler equation are all the external forces acting on the system, namely, due to pressure gradient, due to the gravitational potential ($\psi$) and due to any other external potential ($\Psi$), respectively. The last two terms denote forces due to viscosity.

In our calculations, we assume, $\zeta = 0$. The modified Navier–Stokes equation after making this assumption is given by

$$
\rho(\partial_t \vec{v} + \vec{v} \cdot \nabla \vec{v}) = -\nabla p - \rho \nabla \psi - \rho \nabla \Psi + \eta \nabla^2 \vec{v} + \tfrac{1}{3}\eta \nabla(\nabla \cdot \vec{v}).
$$

Considering the flow to be locally irrotational, i.e., $\nabla \times \vec{v} = 0$, we can assume that the flow velocity $\vec{v}$ has a form, $\vec{v} = -\nabla \phi$, where $\phi$ is the velocity potential. Incorporating this in the Euler and Navier–Stokes equations, we get

$$
\begin{aligned}
\partial_t \rho - \nabla \cdot (\rho \nabla \phi) &= 0 \\
\rho(\partial_t \nabla \phi - \nabla \phi \cdot \nabla(\nabla \phi)) &= \nabla p + \rho \nabla \psi + \rho \nabla \Psi - \eta \nabla^2(\nabla \phi) - \frac{1}{3}\eta \nabla(\nabla \cdot (\nabla \phi))
\end{aligned}
\tag{2}
$$

Following standard vector algebra, Equation (2) can be written as

$$
\nabla\left[-\partial_t \phi + \frac{1}{2}(\nabla \phi)^2 + h + \psi + \frac{4}{3}\nu \nabla^2 \phi\right] = -\frac{4}{3}\nu \nabla \log \rho \nabla^2 \phi
\tag{3}
$$

where $\nu = \frac{\eta}{\rho}$ is the kinematic viscosity and $h$ is the specific enthalpy, given by, $\nabla h = \frac{1}{\rho}\nabla p$. Equation (3) is known as Burgers' equation [12].

### 2.2. Analogue Gravity in Inviscid Fluids

This subsection summarizes the rudiments of Unruh's incipient work on the derivation of an acoustic analogue spacetime with a Lorentzian metric from the acoustic perturbation of a barotrpic, irrotational and inviscid fluid [2,3]. For inviscid systems, the continuity equation remains the same as described in Equation (1), while Burgers' equation becomes

$$
-\partial_t \phi + h + \frac{1}{2}(\nabla \phi)^2 + \psi + \Psi = 0
\tag{4}
$$

This is the Euler equation for the velocity potential $\phi$. Equations (1) and (4) are linearized around the background $(\rho_0, \phi_0)$, using $\rho = \rho_0 + \epsilon \rho_1$ and $\phi = \phi_0 + \epsilon \phi_1$. Here $\rho_1$ and $\phi_1$ are the perturbations caused in the background density and velocity potential, respectively, due to the sonic disturbances.

Linearizing the continuity equation leads to the following equations:

$$
\begin{aligned}
\mathcal{O}(\epsilon^0) &: \partial_t \rho_0 - \vec{\nabla} \cdot (\rho_0 \vec{\nabla} \phi_0) = 0 \tag{5} \\
\mathcal{O}(\epsilon) &: \partial_t \rho_1 - \vec{\nabla} \cdot (\rho_1 \vec{\nabla} \phi_0 + \rho_0 \vec{\nabla} \phi_1) = 0 \tag{6}
\end{aligned}
$$

Using $\nabla h(p) = \frac{1}{\rho}\nabla p$, we can linearize $h$ as $h = h_0 + \epsilon h_1$, where $h_1 = \frac{1}{\rho_0}p_1$. The Euler equation at $\mathcal{O}(\epsilon^0)$ and $\mathcal{O}(\epsilon^1)$ is given by

$$\mathcal{O}(\epsilon^0) \; : \; -\partial_t\phi_0 + h_0 + \frac{1}{2}(\vec{\nabla}\phi_0)^2 + \psi \;\; = \;\; 0 \tag{7}$$

$$\mathcal{O}(\epsilon) \; : \; -\partial_t\phi_1 + \frac{p_1}{\rho_0} + \vec{\nabla}\phi_0 \cdot \vec{\nabla}\phi_1 \;\; = \;\; 0 \tag{8}$$

Substituting $p_1$ from Equation (8) into Equation (6) by using $\rho_1 = (\partial\rho/\partial p)p_1$, we get

$$-\partial_t\left(\frac{\partial\rho}{\partial p}\rho_0(\partial_t\phi_1 - \nabla\phi_0 \cdot \nabla\phi_1)\right) + \nabla \cdot \left(\rho_0\nabla\phi_1 + \rho_0\frac{\partial\rho}{\partial p}\nabla\phi_0(\partial_t\phi_1 - \nabla\phi_0 \cdot \nabla\phi_1)\right) = 0 \tag{9}$$

Equation (9) can be compactly written in the form

$$\partial_\alpha(f^{\alpha\beta}\partial_\beta\phi_1) = 0 \tag{10}$$

where

$$f^{\alpha\beta} = \frac{\rho_0}{c^2}\left[\begin{array}{c|c} -1 & -v_0^i \\ \hline -v_0^j & c^2\delta^{ij} - v_0^i v_0^j \end{array}\right]$$

Here $\frac{1}{c^2} = (\partial\rho/\partial p)_0$ with $c$ being the local speed of sound in the unperturbed fluid.

Now, the equation of motion followed by scalar fields ($\phi_1$) propagating on a Lorentzian spacetime with a metric $g_{\alpha\beta}$ is given by

$$\Delta\phi_1 = \frac{1}{\sqrt{-g}}\partial_\alpha(\sqrt{-g}g^{\alpha\beta}\partial_\beta\phi_1) = 0 \tag{11}$$

where $\Delta\phi_1$ is the d'Alembertian of $\phi_1$ and $g = det(g_{\alpha\beta})$. For Equation (10) to resemble Equation (11), we must have

$$\sqrt{-g}g^{\alpha\beta} = f^{\alpha\beta} \tag{12}$$

Using this equation, we get

$$g^{\alpha\beta} = \frac{1}{\rho_0 c}\left[\begin{array}{c|c} -1 & -v_0^i \\ \hline -v_0^j & c^2\delta^{ij} - v_0^i v_0^j \end{array}\right]$$

Inverting $g^{\alpha\beta}$, we get the acoustic metric

$$g_{\alpha\beta} = \frac{\rho_0}{c}\left[\begin{array}{c|c} -(c^2 - v_0^2) & -v_0^i \\ \hline -v_0^j & \delta_{ij} \end{array}\right]$$

The signature of this metric is Lorentzian (-,+,+,+) and it thus describes the geometry of a Lorentzian acoustic spacetime as seen by the first-order sonic perturbations. It is clear that this metric is dependent on the flow parameters of the inviscid fluid in question, namely, the density and velocity of the unperturbed fluid. For quantum fluids such as superfluid helium or Bose–Einstein condensates in the hydrodynamic approximation, this acoustic metric is the starting point of many an assay to produce experimentally accessible phenomena. As already mentioned, some of these have actually been observed. However, most classical fluids are not inviscid and therefore remain outside the realm of observational accessibility. Nevertheless, Torres et al. [9] broke new ground by beginning experimentation with water, a low-viscosity liquid which is freely available naturally. This raises the question whether one can relax the condition of inviscidity and study viscous fluids to explore possible acoustic geometries under perturbation. To this we now turn.

### 2.3. Violation of Lorentz Invariance Due to Viscosity

Acoustic general relativity retains its local Lorentz invariance akin to the formulation of physical spacetime geometry in general relativity. This is taken to be a hallmark of the entire acoustic gravity analogy. However, Visser [3] has argued that for fluids with viscosity, this local Lorentz invariance may have to be sacrificed, thereby disturbing one of the key

underpinnings of the acoustic analogy per se. In this subsection we summarize this argument, showing how viscosity breaks Lorentz invariance, with an important modification. It has been assumed [3] that the *kinematic* coefficient of viscosity ($\nu$) is spatially constant, so that it remains constant under acoustic perturbations. However, it is well known [11] that the *dynamic* viscosity ($\eta$) remains constant under pressure/density fluctuations. Since $\nu \equiv \eta/\rho$, and under acoustic perturbations $\rho = \rho_0 + \epsilon\rho_1$, we have

$$\nu \simeq \nu_0 \left(1 - \epsilon\frac{\rho_1}{\rho_0}\right) \tag{13}$$

where $\nu_0$ is the kinematic coefficient of viscosity in the absence of acoustic perturbations. We also absorb the $4/3$ factor into $\nu_0$. This coefficient of kinematic viscosity is itself *not* spatially constant, as assumed in [3], if, as stated in [11], the dynamic coefficient of viscosity is under pressure/density perturbations. This calls into question the derivation of Burger's equation for viscous fluids given in [3], whose acoustic perturbations are then studied there. Fortunately, there is a physical argument for which Burger's equation is still valid as an *approximate* equation: the case for small kinematic viscosity $\nu$ and, also, a slowly varying density, such that $\nabla \log\rho_0$ is small. If both these hold, i.e., more precisely, if $|\nu\nabla \log\rho\nabla^2\phi| << |\nabla(\nu\nabla^2\phi)|$, everywhere in the fluid, then indeed Burger's equation holds as an approximate equation:

$$-\partial_t\phi + \frac{1}{2}(\nabla\phi)^2 + h + \psi + \nu\nabla^2\phi \approx 0 \tag{14}$$

We shall assume in this subsection that Equation (14) is obeyed as an exact equality, even though we do not assume that $\nu$ is exactly spatially constant. In other words, the acoustic density perturbation effect in Equation (13) shall indeed be taken into account.

Following the procedure of linearization on Equation (14) as described in the previous section, we have

$$\mathcal{O}(\epsilon^0) : \quad -\partial_t\phi_0 + h_0 + \frac{1}{2}(\vec{\nabla}\phi_0)^2 + \psi + \nu_0\nabla^2\phi_0 = 0 \tag{15}$$

$$\mathcal{O}(\epsilon^1) : \quad -\partial_t\phi_1 + \frac{p_1}{\rho_0} + \vec{\nabla}\phi_0 \cdot \vec{\nabla}\phi_1 + \nu_0(\nabla^2\phi_1 - \frac{\rho_1}{\rho_0}\nabla^2\phi_0) = 0 \tag{16}$$

The linearized continuity equations remain the same as in the inviscid case (Equations (5) and (6)). Using $p_1 = c^2\rho_1$ in Equation (16), we can solve for $\rho_1$, yielding

$$\rho_1 = \frac{\rho_0}{c^2 - \nu_0 D_t \log\rho_0}\left(D_t\phi_1 - \nu_0\nabla^2\phi_1\right),$$

$$\simeq \frac{\rho_0}{c^2}[D_t\phi_1 - \nu_0(\nabla^2\phi_1 - \frac{1}{c^2}D_t \log\rho_0 D_t\phi_1)] \tag{17}$$

where we have assumed $|(\nu_0^2/c^2)D_t \log\rho_0| << 1$ and defined $D_t \equiv \partial_t + \vec{v}_0 \cdot \nabla$. It is obvious that without this restriction of a small $\nu_0$, a correction to the wave equation for the acoustic perturbations in the form of $\partial_\mu[(f^{\mu\nu} + \nu_0 h^{\mu\nu})\partial_\nu\phi_1] = 0$ is impossible to extract, since the correction term $h^{\mu\nu}$ itself is a function of $\nu_0$. Thus, in this form it is not easy to extract an effective velocity-dependent spatial correction metric, as has been obtained in [3]. However, with this restriction, some progress towards showing Lorentz violation can be made.

If we ignore the non-constancy of $\nu$ or its change under density perturbations, i.e., follow [3], the continuity equation under acoustic perturbations (6) can be used with (17) to obtain

$$\partial_\alpha(f^{\alpha\beta}\partial_\beta\phi_1) = -\rho_0\nu(\partial_t - \nabla\phi_0 \cdot \nabla)(\frac{1}{c^2}\nabla^2\phi_1) \tag{18}$$

Writing the *lhs* of the equation using the d'Alembertian for the inviscid acoustic metric, we get

$$\Delta\phi_1 = -\frac{\nu c}{\rho_0}(\partial_t - \nabla\phi_0 \cdot \nabla)(\frac{1}{c^2}\nabla^2\phi_1) \tag{19}$$

Defining the fluid 4-velocity, $V^\alpha = \frac{1}{\sqrt{\rho_0 c}}(1, v_0^x, v_0^y, v_0^z)$, and a modified metric with the modification only having a spatial part $g_{space}$, Equation (17) can be written as

$$\Delta\phi_1 = -\frac{\nu c^2}{\sqrt{\rho_0}}V^\alpha\partial_\alpha(\frac{1}{c^2}\nabla^2\phi_1) \tag{20}$$

The Laplacian in the above equation is expressible only in terms of the spatial metric $g_{space}^{\alpha\beta}$.

The acoustic metric can be written in terms of the spatial metric and the fluid 4-velocity as

$$g^{\alpha\beta} = -V^\alpha V^\beta + \frac{c}{\rho_0}g_{space}^{\alpha\beta} \tag{21}$$

The presence of $V^\alpha$ in this equation breaks the Lorentz invariance as all inertial frames are no longer equivalent. Thus, the direct introduction of viscosity leads to the violation of Lorentz symmetry. Inclusion of the non-constancy effects of the viscosity complicates the extraction of a modified spatial metric since terms $O(D_t\nu)$ will have to be taken into account.

It is not an easy proposition that an acoustic metric à la Unruh [2] be envisaged from this formulation. Under the circumstance, perhaps a new strategy must be evolved to deal with fluids with a small viscosity. It appears logical to adopt a method of *double* perturbations, where we perturb the fluid potentials (and hence the velocity field) in terms of *both* acoustic perturbations and viscosity. Clearly, the fluid density/pressure are exempt from the viscosity perturbations, since these, in some sense, are intrinsic parameters of the fluid, and hence independent of the viscosity coefficient. In any case, this is the assumption we make here. It will turn out that, with this assumption, up to the first order in both perturbations, the sequence in which the perturbations are introduced does not make any difference. Our perturbations equations will indeed reduce to Equation (18) in some approximation. However, simplifications will result from our approach over and above that of [3], as we show in the next section.

## 3. Double Perturbation

### 3.1. The Formulation

To investigate slightly viscous fluid systems, we consider viscosity as perturbations and change the flow parameters of the fluid accordingly. Upon introducing first-order sonic and viscosity perturbations into an inviscid system, around the equilibrium state of the background $(\phi_0, \rho_0)$, one must remember that $\nu_0$ is not a constant because of its dependence on the density. This implies that there will be additional terms given by the spacetime gradient $\partial_\mu\nu_0 = -\nu_0\partial_\mu\log\rho_0$ where $\mu = t, x, y, z$. The perturbed density and velocity potential are given by

$$\rho = \rho_0 + \epsilon\rho_1 \tag{22}$$

$$\nu = \nu_0\left(1 - \epsilon\frac{\rho_1}{\rho_0}\right) \tag{23}$$

$$\phi = \phi_{0I} + \epsilon\phi_{1I} + \nu_0\phi_{0V} + \epsilon\nu_0\phi_{1V} \tag{24}$$

$$\partial_\mu\phi = \partial_\mu\phi_{0I} + \epsilon\partial_\mu\phi_{1I} + \nu_0(\partial_\mu\phi_{0V} - \phi_{0V}\partial_\mu\log\rho_0)$$
$$+ \epsilon\nu_0(\partial_\mu\phi_{1V} - \phi_{1V}\partial_\mu\log\rho_0) \tag{25}$$

where $\phi_{0I}$ and $\rho_0$ are the background velocity potential and background density, respectively, $\phi_{1I}$ and $\rho_1$ are the acoustic perturbation on $\phi_{0I}$ and $\rho_{0I}$, respectively, $\phi_{0V}$ is the viscosity perturbation on $\phi_{0I}$ and $\phi_{1V}$ is the viscosity perturbation on $\phi_{1I}$. Observe that $\phi_{0V}$

and $\phi_{1V}$ are dimensionless quantities, since the kinematic viscosity coefficient $\nu_0$ carries the dimension of the velocity potential.

Going back to the derivation of Burger's equation as per [3], we realize that we must replace (14) by

$$\nabla[-\partial_t\phi + \frac{1}{2}(\nabla\phi)^2 + h + \psi + \nu\nabla^2\phi] = -\nu\nabla\log\rho\nabla^2\phi \tag{26}$$

Defining the total derivative $D_t \equiv \partial_t + \vec{v}_{0I} \cdot \nabla$, the doubly perturbed continuity and Euler–Navier–Stokes equations derived from (26), at different orders of perturbation, are, respectively, as follows:

$\mathcal{O}(\epsilon^0 \nu_0^0)$:

$$\partial_t\rho_0 - \nabla \cdot (\rho_0\nabla\phi_{0I}) = 0 \tag{27}$$

$$-\partial_t\phi_{0I} + h_0 + \frac{1}{2}(\nabla\phi_{0I})^2 + \psi = 0 \tag{28}$$

$\mathcal{O}(\epsilon\nu_0^0)$:

$$D_t\rho_1 - \rho_1 D_t\log\rho_0 - \nabla \cdot (\rho_0\nabla\phi_{1V}) = 0 \tag{29}$$

$$-D_t\phi_{1I} + \frac{c^2\rho_1}{\rho_0} = 0 \tag{30}$$

$\mathcal{O}(\epsilon^0\nu_0)$:

$$\nabla \cdot (\rho_0\nabla\phi_{0V} - \phi_{0V}\nabla\rho_0) = 0 \tag{31}$$

$$\nabla[-D_t\phi_{0V} + \phi_{0V}D_t\log\rho_0 + \nabla^2\phi_{0I}] = -\nabla\log\rho_0\nabla^2\phi_{0I} \tag{32}$$

$\mathcal{O}(\epsilon\nu_0)$:

$$\nabla \cdot [\rho_1\nabla\phi_{0V} + \rho_0\nabla\phi_{1V} - (\rho_1\phi_{0V} + \rho_0\phi_{1V})\nabla\log\rho_0] = 0 \tag{33}$$

$$\nabla[-D_t\phi_{1V} + \phi_{1V}D_t\log\rho_0 \quad + \quad \nabla^2\phi_{1I} + \nabla\phi_{1I} \cdot (\nabla\phi_{0V} - \phi_{0V}\nabla\log\rho_0) - \frac{\rho_1}{\rho_0}\nabla^2\phi_{0I}]$$
$$= \quad -\nabla\log\rho_0\nabla^2\phi_{1I} - \nabla^2\phi_{0I}\nabla(\frac{\rho_1}{\rho_0}) + \frac{\rho_1}{\rho_0}\nabla\log\rho_0\nabla^2\phi_{0I} \tag{34}$$

These are the complete first-order acoustic and viscosity perturbation equations resulting from the continuity and Navier–Stokes equations. We retain terms of $\mathcal{O}(\epsilon\nu_0)$ since these terms are first order in the two small parameters $\epsilon$ and $\nu_0$, which are physically quite distinct from each other and cannot be regarded as of the same order of magnitude numerically.

Let us first verify that the inviscid case behaviour has been reproduced. Substituting $p_1$ from Equation (30) into Equation (29) by using $\rho_1 = p_1/c^2$, we get

$$-\partial_t(\frac{\rho_0}{c^2}(\partial_t\phi_{1I} + \vec{v}_0 \cdot \nabla\phi_{1I})) + \nabla \cdot (\rho_0\nabla\phi_{1I} - \frac{\rho_0}{c^2}\vec{v}_0(\partial_t\phi_{1I} + \vec{v}_0 \cdot \nabla\phi_{1I})) = 0 \tag{35}$$

This equation can also be written as

$$-\rho_0 D_t(\frac{1}{c^2}D_t\phi_{1I}) + \nabla \cdot (\rho_0\nabla\phi_{1I}) = 0 \tag{36}$$

Equation (35) is the same equation that is obtained in the inviscid case (Equation (9)) for first-order acoustic perturbations and can thus be compactly written in the form

$$\partial_\alpha(f^{\alpha\beta}\partial_\beta\phi_{1I}) = 0 \tag{37}$$

with $f^{\alpha\beta}$ being given in Section 2.2.

*3.2. Consequences: Slowly Varying Background Density*

From the $O(\epsilon^0 v_0^0)$ continuity equation (27) we obtain

$$\nabla \cdot \vec{v}_{0I} \equiv -\nabla^2 \phi_{0I} = -D_t \log \rho_0 \tag{38}$$

where $D_t = \partial_t + \vec{v}_{0I} \cdot \nabla$, $\vec{v}_{0I} = -\nabla \phi_{0I}$. Similarly, from Equation (30), using the definition $h_1 = c^2 \rho_1 / \rho_0$, we get

$$\rho_1 = \frac{\rho_0}{c^2} D_t \phi_{1I} . \tag{39}$$

As expected, Equation (39) considerably simplifies the complicated non-linear dependence of $\rho_1$ on the kinematic viscosity coefficient $v_0$ observed in Equation (17). Equation (39) is of course the same relation obtained in the inviscid case.

It is clear that second-order derivatives and terms of second degree in the first derivative of $\log \rho_0$ appear in the continuity and Euler–Navier–Stokes equations above. If we assume that $\log \rho_0$ is sufficiently slowly varying, we can ignore these second-order and second-degree terms, as we shall do from now on. Solving the continuity equation (31), we get

$$\rho_0 \nabla^2 \phi_{0V} - \phi_{0V} \nabla^2 \rho_0 = 0 \tag{40}$$

By inspection, in view of the the Navier–Stokes equation (32) at this order of perturbation theory, a solution of $\phi_{0V}$ is given by

$$\phi_{0V} = \frac{\rho_0}{\bar{\rho}_0} + \log \rho_0 \tag{41}$$

where, $\bar{\rho}_0$ is an arbitrary dimensional constant. Substituting this in Equation (34), one obtains

$$\nabla \left[ -D_t \phi_{1V} + \phi_{1V} D_t \log \rho_0 + \nabla^2 \phi_{1I} - \frac{D_t \phi_{1I}}{c^2} D_t \log \rho_0 + \nabla \log \rho_0 \nabla \phi_{1I} \right] = 0 \tag{42}$$

Using Equation (36) in the above equation and solving, we can cast Equation (42) into

$$(D_t - D_t \log \rho_0)(\phi_{1V} - \frac{1}{c^2} D_t \phi_{1I}) = 0 \tag{43}$$

This equation is satisfied by

$$\phi_{1V} = \frac{1}{c^2} D_t \phi_{1I} + \frac{\rho_0}{\bar{\rho}_1} \tag{44}$$

where $\bar{\rho}_1$ is an arbitrary dimensional constant. Substituting this solution in Equation (33), we get

$$\nabla^2 (\frac{1}{c^2} D_t \phi_{1I}) = (\nabla^2 \rho_0 / \rho_0) \frac{1}{c^2} D_t \phi_{1I} \tag{45}$$

Since $(\nabla^2 \rho_0 / \rho_0) = \nabla^2 \log \rho_0 + (\nabla \log \rho_0)^2$, the RHS of Equation (45) can be neglected. Thus, we get the constraint equation:

$$\nabla^2 (\frac{1}{c^2} D_t \phi_{1I}) \approx 0 \tag{46}$$

## 4. Aspects of Slightly Viscous Acoustic Geometry

*4.1. Perturbed $f^{\mu\nu}$ Tensor*

We are treating the viscosity of the system as a perturbation on its various flow parameters. To explore the possibility of an acoustic geometry for this slightly viscous system, we can perturb the $f^{\alpha\beta}$ tensor and calculate the perturbation correction to the $f^{\alpha\beta}$ tensor, namely $h^{\alpha\beta}$. This formalism allows us to write the equation of motion of sound waves in slightly viscous fluids systems as

$$\partial_\alpha((f^{\alpha\beta} - \nu_0 h^{\alpha\beta})\partial_\beta(\phi_{1I} + \nu_0\phi_{1V})) = 0 \tag{47}$$

Linearizing Equation (47) w.r.t. $\nu_0$, we have
$\mathcal{O}(\nu_0^0)$:

$$\partial_\alpha(f^{\alpha\beta}\partial_\beta\phi_{1I}) = 0 \tag{48}$$

$\mathcal{O}(\nu_0)$:

$$
\begin{aligned}
\partial_\alpha(f^{\alpha\beta}\partial_\beta\phi_{1V}) - \partial_\alpha(\phi_{1V}f^{\alpha\beta}\partial_\beta\log\rho_0) \quad & - \quad \partial_\alpha\log\rho_0 f^{\alpha\beta}\partial_\beta\phi_{1V} + \phi_{1V}f^{\alpha\beta}\partial_\alpha\log\rho_0\partial_\beta\log\rho_0 \\
& = \quad \partial_\alpha(h^{\alpha\beta}\partial_\beta\phi_{1I}) - \partial_\alpha\log\rho_0 h^{\alpha\beta}\partial_\beta\phi_{1I}
\end{aligned}
\tag{49}
$$

Equation (48) is the same as Equation (37). Its existence at $O(\nu_0^0)$ makes sense as this is the equation of first-order acoustic perturbations in inviscid systems.

Substituting $\phi_{1V} = \frac{1}{c^2}D_t\phi_{1I} + (\rho_0/\bar{\rho}_1)$ in Equation (49), we get

$$
\begin{aligned}
& \partial_\alpha(f^{\alpha\beta}\partial_\beta(\frac{1}{c^2}D_t\phi_{1I})) - \partial_\alpha(\frac{1}{c^2}D_t\phi_{1I}f^{\alpha\beta}\partial_\beta\log\rho_0) - \partial_\alpha\log\rho_0 f^{\alpha\beta}\partial_\beta(\frac{1}{c^2}D_t\phi_{1I}) \\
& + \quad \frac{1}{c^2}D_t\phi_{1I}f^{\alpha\beta}\partial_\alpha\log\rho_0\partial_\beta\log\rho_0 = \partial_\alpha(h^{\alpha\beta}\partial_\beta\phi_{1I}) - \partial_\alpha\log\rho_0 h^{\alpha\beta}\partial_\beta\phi_{1I}
\end{aligned}
\tag{50}
$$

We get Equation (50) at $\mathcal{O}(\epsilon\nu_0)$. At $\mathcal{O}(\epsilon^0\nu)$, we considered the approximation of neglecting second- and higher-order derivatives of $\log\rho_0$. At $\mathcal{O}(\epsilon\nu_0)$ it would be reasonable to neglect first- and higher-order derivatives of $\log\rho_0$. Since $f^{\alpha\beta}$ is proportional to $\rho_0$ (Section 2.2), we divided Equation (50) with $\rho_0$ and apply the above approximation to get

$$\frac{1}{\rho_0}\partial_\alpha(f^{\alpha\beta}\partial_\beta\frac{1}{c^2}D_t\phi_{1I}) = \partial_\alpha(\frac{1}{\rho_0}h^{\alpha\beta}\partial_\beta\phi_{1I}) \tag{51}$$

Similar to Equation (36), we can rewrite the LHS of Equation (51) to obtain

$$-D_t(\frac{1}{c^2}D_t(\frac{1}{c^2}D_t\phi_{1I})) + \frac{1}{\rho_0}\nabla\cdot(\rho_0\nabla(\frac{1}{c^2}D_t\phi_{1I})) = \partial_\alpha(\frac{1}{\rho_0}h^{\alpha\beta}\partial_\beta\phi_{1I}) \tag{52}$$

Using Equation (36) in Equation (52) and solving, we get

$$
\begin{aligned}
2\nabla\frac{1}{c^2}\cdot\nabla D_t\phi_{1I} + \frac{1}{c^2}\nabla\phi_{1I}\cdot\nabla\nabla^2\phi_{0I} - \frac{2}{c^2}\nabla\cdot((\nabla\phi_{1I}\cdot\nabla)\nabla\phi_{0I}) + D_t\phi_{1I}\nabla^2\frac{1}{c^2} \\
- D_t\frac{1}{c^2}\nabla^2\phi_{1I} = \partial_\alpha(\frac{1}{\rho_0}h^{\alpha\beta}\partial_\beta\phi_{1I})
\end{aligned}
$$

The components of $h^{\alpha\beta}$ are determined by comparing the coefficients of the second-order derivatives of $\phi_{1I}$ on both sides of the above equation. These components are then verified by comparing the coefficients of first-order derivatives of $\phi_{1I}$. The perturbation correction tensor, $h^{\alpha\beta}$, thus calculated is given by

$$h^{\alpha\beta} = \left[\begin{array}{c|c} 0 & \rho_0\partial_i\frac{1}{c^2} \\ \hline \rho_0\partial_j\frac{1}{c^2} & (-\rho_0 D_t\frac{1}{c^2})\delta_{ij} - \rho_0\partial_i\frac{1}{c^2}\partial_j\phi_{0I} - \rho_0\partial_j\frac{1}{c^2}\partial_i\phi_{0I} - \frac{2}{c^2}\rho_0\partial_i\partial_j\phi_{0I} \end{array}\right]$$

### 4.2. The Perturbed Acoustic Metric

The equation of motion followed by scalar fields ($\phi_1$) propagating on a Lorentzian background with a metric $g_{\alpha\beta P}$ is

$$\frac{1}{\sqrt{-g_P}}\partial_\alpha(\sqrt{-g_P}g_P^{\alpha\beta}\partial_\beta\phi_1) = 0 \tag{53}$$

In our case, for the equation of motion of sonic perturbations (Equation (47)) to resemble Equation (53), we must have, $\phi_1 = \phi_{1I} + \nu_0\phi_{1V}$, and

$$\sqrt{-g_P}\,g_P^{\alpha\beta} = f^{\alpha\beta} - \nu_0 h^{\alpha\beta} \tag{54}$$

Taking the determinant on both sides and solving, we get

$$\sqrt{-g_P} = \sqrt{-det(f^{\alpha\beta})}(1 - \frac{\nu_0}{2}f_{\alpha\beta}h^{\alpha\beta}) \tag{55}$$

Further solving, we get

$$\sqrt{-g_P} = \frac{\rho_0^2}{c}(1 + \nu_0(D_t\frac{3}{2c^2} + \frac{1}{c^2}\nabla^2\phi_{0I})) \tag{56}$$

Using Equation (54), we get the inverse metric $g_P^{\alpha\beta}$ as

$$g_P^{\alpha\beta} = \frac{1-\nu_0 k}{\rho_0 c}$$

$$\left[\begin{array}{c:c} -1 & -v_{0I}^i + \frac{2}{c}\nu_0\partial_i c \\ \hdashline -v_{0I}^j + \frac{2}{c}\nu_0\partial_j c & (c^2 - \frac{2}{c}\nu_0 D_t c)\delta^{ij} - v_{0I}^i v_{0I}^j + \frac{2}{c}\nu_0 v_{0I}^j\partial_i c + \frac{4}{c}\nu_0 v_{0I}^i\partial_j c - 2\nu\partial_i v_{0I}^j \end{array}\right]$$

where, $k = D_t\frac{3}{2c^2} + \frac{1}{c^2}\nabla^2\phi_{0I}$, $\vec{v}_{0I} = -\nabla\phi_{0I}$ is the background velocity of the fluid and $v_{0I}^i$ are its spacial components.

The acoustic metric, obtained by inverting $g_P^{\alpha\beta}$, is given by

$$g_{\alpha\beta P} = \frac{(1-\nu_0 k)\rho_0}{c}$$

$$\left[\begin{array}{c:c} \begin{array}{c} -[(c^2 - v_{0I}^2)(1 + 2k\nu_0) + 4\nu_0(\vec{v}_{0I}\cdot\nabla\log c) \\ -\frac{2}{c^2}\nu_0 v_{0I}^2 D_t\log c - \frac{2}{c^2}\nu_0\vec{v}_{0I}\cdot((\vec{v}_{0I}\cdot\nabla)\vec{v}_{0I})] \end{array} & \begin{array}{c} -v_{0I}^i(1 + 2k\nu_0) + 2\nu_0\partial_i\log c \\ -\frac{2}{c^2}\nu_0(v_{0I}^i D_t\log c + (\vec{v}_{0I}\cdot\nabla)v_{0I}^i) \end{array} \\ \hdashline \begin{array}{c} -v_{0I}^j(1 + 2k\nu_0) + 2\nu_0\partial_j\log c \\ -\frac{2}{c^2}\nu_0(v_{0I}^j D_t\log c + (\vec{v}_{0I}\cdot\nabla)v_{0I}^j) \end{array} & \begin{array}{c} \delta_{ij}(1 + 2k\nu_0) + \frac{2}{c^2}\nu_0 D_t\log c \\ +\frac{2}{c^2}\nu_0\partial_i v_{0I}^j \end{array} \end{array}\right]$$

### *4.3. (2+1)-Dimensional Vortex-Type Flow*

The application of the above-mentioned concepts in vortex geometries results in much simplification as the flow is now restricted to only two spacial dimensions instead of three as was described in the general case. The continuity equation and the vorticity free constraint, along with the conservation of angular momentum, leads to the background density $\rho_0$ to be position independent. This also leads to a position-independent background pressure due to the barotropicity of the fluid [3]. These two conditions together make the speed of sound ($c$) spatially constant as well. Since the flow is non-turbulent, we also considered $\rho_0$ and thus $c$ to be constant in time. These conditions imply that $k = 0$ and $D_t\log c = \partial_i\log c = 0$. The above-mentioned simplifications lead to the following acoustic metric:

$$g_{\alpha\beta P} = \frac{\rho_0}{c}\left[\begin{array}{c:c} -(c^2 - v_{0I}^2) + \frac{2}{c^2}\nu_0\vec{v}_{0I}\cdot[(\vec{v}_{0I}\cdot\nabla)\vec{v}_{0I}] & -v_{0I}^i - \frac{2}{c^2}\nu_0(\vec{v}_{0I}\cdot\nabla)v_{0I}^i \\ \hdashline -v_{0I}^j - \frac{2}{c^2}\nu_0(\vec{v}_{0I}\cdot\nabla)v_{0I}^j & \delta_{ij} + \frac{2}{c^2}\nu_0\partial_i v_{0I}^j \end{array}\right]$$

As is evident, this metric is significantly less cluttered than the one in the general case, which makes it much easier to work with.

## 5. Discussion

We have thus gleaned from the doubly perturbed irrotational fluid mechanics equations a geometrical structure as a generalization of that discerned by Unruh [2] for inviscid fluids, to the case of fluids with a small kinematic viscosity. A perturbed acoustic metric, seen by linear acoustic perturbations, was also derived in Section 4.3. What has not been investigated in the foregoing subsections is the *signature* of the perturbed metric. However, unlike the acoustic metric derived for the inviscid case, here the perturbations most likely

change the signature away from the Lorentzian structure that the inviscid case exhibited. We believe that this change in the signature of the acoustic metric under viscosity effects, albeit small, is precisely as it should be. Viscosity heralds dissipation into an otherwise non-dissipative system, which must underlie the discovery of the Lorentzian acoustic geometry in the inviscid case. With dissipation present, a pristine Lorentzian-perturbed acoustic geometry would be nothing short of a miracle. The description of fluids with dissipation described as a macroscopic viscous flow, with microscopic physics details coarse-grained over, must automatically break the local Lorentz invariance of the inviscid acoustic geometry.

The situation with viscosity in fluids is in some ways reminiscent of Lorentz-invariance violation in the electrodynamics of material media modelled as a macroscopic continuum with some specific properties of permittivity and permeability [13]. The constitutive relation between the electric and magnetic fields and their *excitations*, namely, the displacement field and the magnetic induction field, whenever non-trivial, automatically signifies a breakdown of Lorentz symmetry in the underlying spacetime. This happens because of the coarse-grained continuum approximation of the material medium, which in reality is atomistic in nature and therefore *not* a continuum. Coarse-grained continuum description of matter necessarily breaks spacetime symmetries of vacua. From this standpoint, this Lorentz violation in the acoustic geometry in viscous fluids is expected.

We end this paper with a mention of the pending issues which we have not addressed in this essay. First of all, the restriction of our formulation to *small* viscosity was motivated by ongoing experiments with water, which indeed has a small coefficient of kinematic viscosity at laboratory temperature. This is therefore far from general. The technical simplicity of this restriction enabled the use of the double perturbation scheme which in turn led us to the viscosity-perturbed acoustic geometry. One certainly needs to go beyond this approximation if one is to have a large array of fluids to deal with. Similarly, the restriction to fluids with spatially slowly varying density enabled us to set up an approximation where derivatives of the logarithm of the density could be ignored. The acoustic geometry of fluids whose densities change rapidly over space is thus not captured by our approach. Hopefully, though, our approach has expanded the scope, albeit in a small way, of the acoustic analogue geometrical description of sonic perturbations in fluids beyond the incipient works.

**Author Contributions:** Conceptualization: P.M.; Methodology: P.M.; Validation: M.P. and P.M.; Formal Analysis: M.P. and P.M.; Investigation: M.P.; Writing (original draft): M.P.; Writing (Review and Editing): M.P. and P.M.; Supervision: P.M. All authors have read and agreed to the published version of the manuscript.

**Funding:** No funding reeived from any sources for this research.

**Institutional Review Board Statement:** No Institutional Review has been conducted for this work.

**Informed Consent Statement:** Not applicable for studies not involving humans.

**Data Availability Statement:** No data sets have been used in this formal mathematical analysis. All relevant prior references have been duly cited.

**Acknowledgments:** The authors gratefully acknowledge the collaboration and assistance of Suneeta Varadarajan in the initial stages of this work, and for many useful discussions. They also thank O. Ganguly for interesting discussions at the starting phase of the work.

**Conflicts of Interest:** The authors declare no conflict of interest.

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
