# Peer review of "Towards an Acoustic Geometry in Slightly Viscous Fluids"

_universe, doi:10.3390/universe8040205_

Round 1

Reviewer 1 Report

This manuscript can be accepted as is

Author Response

Reviewer 1 has kindly agreed to publish the paper as it is. The authors have no comments to add on this. 

Reviewer 2 Report

See report.

Author Response

  1. Without the assumption of irrotationality, the description of fluid mechanics in terms of velocity potentials is unavailable. This description is crucial to the incipient acoustic analogue gravity result derived in 1981 by W Unruh (Ref. 1), where the acoustically perturbed velocity potential behaves like a scalar field in a Lorentzian curved acoustic geometry. As M Visser has clearly demonstrated (Ref. 2 in the paper), it is possible to generalize this formalism in terms of potentials to the case of viscous fluids. So, viscosity can exist without sacrificing the assumption of irrotationality.

2.   If we change $\nu$ to $nu_0$ in eq. (14), we lose terms when we linearize it at  $O(\epsilon)$. It is because of this the above change cannot be made.

3. Our approach involves two small parameters, the kinematical viscosity and the acoustic perturbation parameter. Even without any acoustic perturbation, because of our restriction to small viscosity, the unperturbed and viscosity perturbed potentials, in absence of any acoustic perturbation, are still distinct in our approach, corresponding to distinct powers of the viscosity coefficient. Thus the identification proposed by Reviewer 2 cannot be made within this `double-perturbation' approach. 

Reviewer 3 Report

This paper deals with the acoustic analogy of gravity. It is a purely theoretical work, and I have no objection to the publication.

I would have seen some link with astrophysical/cosmological observations or, at least, some proposal, so to verify the theory. The authors referred to laboratory experiments aimed at simulating this analogy, but it is exactly an analogy. I suggest to underline that these observations were indeed from laboratory experiments, not from astrophysical/cosmological observations (e.g. page 2, row 40). This is not a little detail, because moving from the laboratory to the large scale structure of the universe implies also a significant change of the impact of gravity. Therefore, those laboratory experiments might be nothing more than just fun toys. Analogies might be useful to drive thoughts, but one must remind that are just analogies. Some parallelism does not necessarily imply a full agreement and suitable description. 

Some typos:

  • page 7, row 183: ... metric à la Unruh...
  • page 7, row 190: ... assumption, up to...
  • page 12, row 332: ... this essay.

Author Response

We thank the Reviewer 3 for point us some typos which have been duly corrected. 

Round 2

Reviewer 2 Report

The authors' response is satisfactory. The paper seems to be publishable in its present form.

Reviewer 3 Report

The revised version is fine for me.